# Environmental Sustainability in Infrastructure Construction—A Review Study on Australian Higher Education Program Offerings

Malindu Sandanayake, Yanni Bouras and Zora Vrcelj *

Institute of Sustainable Industries and Liveable Cities, Victoria University, Melbourne, VIC 3011, Australia
* Correspondence: zora.vrcelj@vu.edu.au

**Abstract:** Infrastructure advancement is a key attribute that defines the development and effective growth of a city or region. Since the introduction of the United Nations Sustainability Development Goals (UN SDGs), more construction companies are focusing on adopting sustainable construction practices. However, a lack of relevant competencies among employees at various infrastructure construction organizations often hinders the successful implementation of sustainable practices. Education that facilitates systematic professional development and contemporary competencies' acquisition is a key to overcoming this barrier. Thus, the current study adopts a three-stage review to identify current research trends and inform future research directions for the enhancement of the environmental sustainability competencies base for infrastructure professionals. A bibliometric assessment was first conducted followed by a focused literature review on sustainability education. Subsequently, two engineering and construction higher education curricula were assessed for infrastructure sustainability content. The results from the three-step analysis indicate that the growing interest in sustainability concepts in the construction industry is driven by policy changes. A lack of financial incentives, the unavailability of resources, a lack of motivation amongst graduates, and limited time in the infrastructure construction sector were identified as some of the major impediments for developing the environmental sustainability competencies base. The requirement for integrated and structured Continuous Professional Development (CPD) programs to facilitate ongoing knowledge acquisition and structured evaluation of professional knowledge in addition to effective undergraduate program development are highlighted. The necessity for a digitally personalised platform that can graphically represent current progress and future milestones and enable peer interaction and collaboration was also identified as critical for improving the uptake of such programs. The findings from this study could be useful for government agencies and infrastructure construction organizations keen to enhance the environmental sustainability knowledge of their employees. Future studies are required to assess sustainability education across the globe and to develop new learning components of infrastructure sustainability that are validated through stakeholder participation.

**Keywords:** Australian construction industry; Continuous Professional Development (CPD); environmental sustainability; higher education; infrastructure

## 1. Introduction

Due to the adverse impacts that conventional construction has on the environment, it is essential that the construction industry transitions to green building practices to achieve national sustainability targets. Since the introduction of the UN Sustainable Development Goals (SDGs) in 2015, governments across the globe are making enormous efforts to promote sustainable practices. Sustainable construction in the infrastructure sector is a key contributor to the responsible consumption of resources without compromising the future environmental, economic, and social benefits within the life cycle of the asset [1–3]. While social and economic impacts are acknowledged for their importance, often, environmental impacts are the major focus when sustainability is considered. The integration of

environmentally sustainable concepts with an infrastructure project is usually achieved through the assistance of rating tools developed by external organisations. For instance, Leadership in Energy and Environmental Design (LEED), the Building Research Establishment Environmental Assessment Method (BREEAM), and the Infrastructure Sustainability (IS) rating tool developed by Infrastructure Sustainability Council (ISC) in Australia provide systematic guidelines for improving the sustainability of construction projects [4–6]. Moreover, addressing knowledge gaps and promoting a positive attitude towards the implementation of sustainable practices in infrastructure projects are the key sustainability drivers in the infrastructure construction sector. Fresh graduates and early-career industry professionals lack competencies and experience to adequately contribute to sustainable infrastructure construction projects. This results in industries allocating additional time and funding to upskill their employees. The rapid pace of infrastructure construction requires industries and governments to seek skilled graduates and industry professionals with suitable sustainability competencies. Architectural, Engineering, and Construction (AEC) curricula in the Australian Higher Education (HE) sector provide numerous opportunities for their graduates to acquire sustainability competencies relevant to the construction industry [7]. There is a contemporary requirement for a structured approach that would bridge the existing competency gaps in education for environmental sustainability in the infrastructure construction industry [8]. Furthermore, the sector often relies on university graduates to lead green construction initiatives based on the competencies they obtained during their university years [9]. To gain a better understanding of the effectiveness of HE studies in preparing students for such roles and the employers' expectations of the graduate competencies level, further research is needed. For Australia, this initiative would enhance competency levels of the future workforce and eventually address the extensive demand for employees with the right skills, knowledge, and attitudes in infrastructure construction.

Existing research on sustainability education for Australian construction and engineering programs has mostly been focused on the building sector. Research on the subject relating to the infrastructure sector is however lacking. The current study aims to address this limitation with the open literature by conducting a systematic review on education for sustainability in the construction industry with a focus on infrastructure projects. The outcomes of the study will inform future directions and advances and will help identify potential barriers for the successful implementation of effective sustainability education for HE students and early-career construction professionals in the Australian infrastructure construction industry. Improving sustainability education is a crucial step in the transition towards a more sustainable construction industry, where development has been inhibited by a lack of understanding and acceptance of sustainable practices amongst construction professionals, in addition to insufficient HE and professional development opportunities [10]. A high level of expertise is necessary for the implementation of sustainable practices, and university education and training play an essential role in knowledge development [11].

## 2. Research Methodology

The main objective of this study was to review current findings and advances related to sustainability education in the infrastructure construction sector. Therefore, a three-step methodology, as shown in Figure 1, was developed to facilitate the gap analysis and to obtain relevant findings. Two review methods, including a bibliometric assessment and a review of the current construction management and civil engineering undergraduate curricula at Victoria University, Australia, were adopted to obtain a general understanding of current environmental sustainability levels in the HE sector. The initial (first) step involved reviewing current research practices and trends using a bibliometric assessment. Bibliometric assessment helps to categorise related data into relevant clusters and groups to facilitate the understanding of the current trends and developments of the selected theme. It is a popular statistical approach that can develop the interconnections and networks using indicators such as keywords, research topics, authors, and publishers of previously

published articles to critically analyse current trends and knowledge gaps [12]. Scopus and the Web of Science (WoS) are the two prominent research databases that can comprehensively capture published literature in the field of construction and engineering. Therefore, the current study used the Scopus search engine to capture relevant previous studies due to its easy adaptability to many bibliometric assessment software packages [13–15]. As shown in Figure 1, the search criteria consisted of research publications including journal articles, conference publications, books, and book chapters from 2000 to 2021. The search string used terms such as "environment*" "sustainab*", "education", "construction", "engineering", and "architect*" using the relevant "AND" or "OR" operator. The asterisk (*) symbol was used to capture all similar keywords and studies on environmental sustainability education in the construction industry. The next (second) step of the methodology involved reviewing academic literature related to the gaps and observations highlighted from the initial step, i.e., bibliometric assessment. The review focused on three subtopics related to learning and teaching infrastructure sustainability including: (i) barriers to sustainable construction practices; (ii) environmental sustainability in built environment programs; and (iii) Continuous Professional Development (CPD).

The third step involved a detailed review of previous research studies and a review of construction management and civil engineering curricula at Victoria University, Melbourne, Australia, to understand the current silos and impediments linked to environmental sustainability education in infrastructure construction. Units in "construction management" and "civil engineering" curricula were reviewed using key words in course-level learning outcomes and unit-level learning outcomes. "Unit" here refers to a subject, and infrastructure-focused or sustainability-focused units in construction management and civil engineering curricula were selected for the review analysis. The findings from the gap analysis were then used to interpret the results and review the follow-up discussions with industry stakeholders undertaken as part of this study.

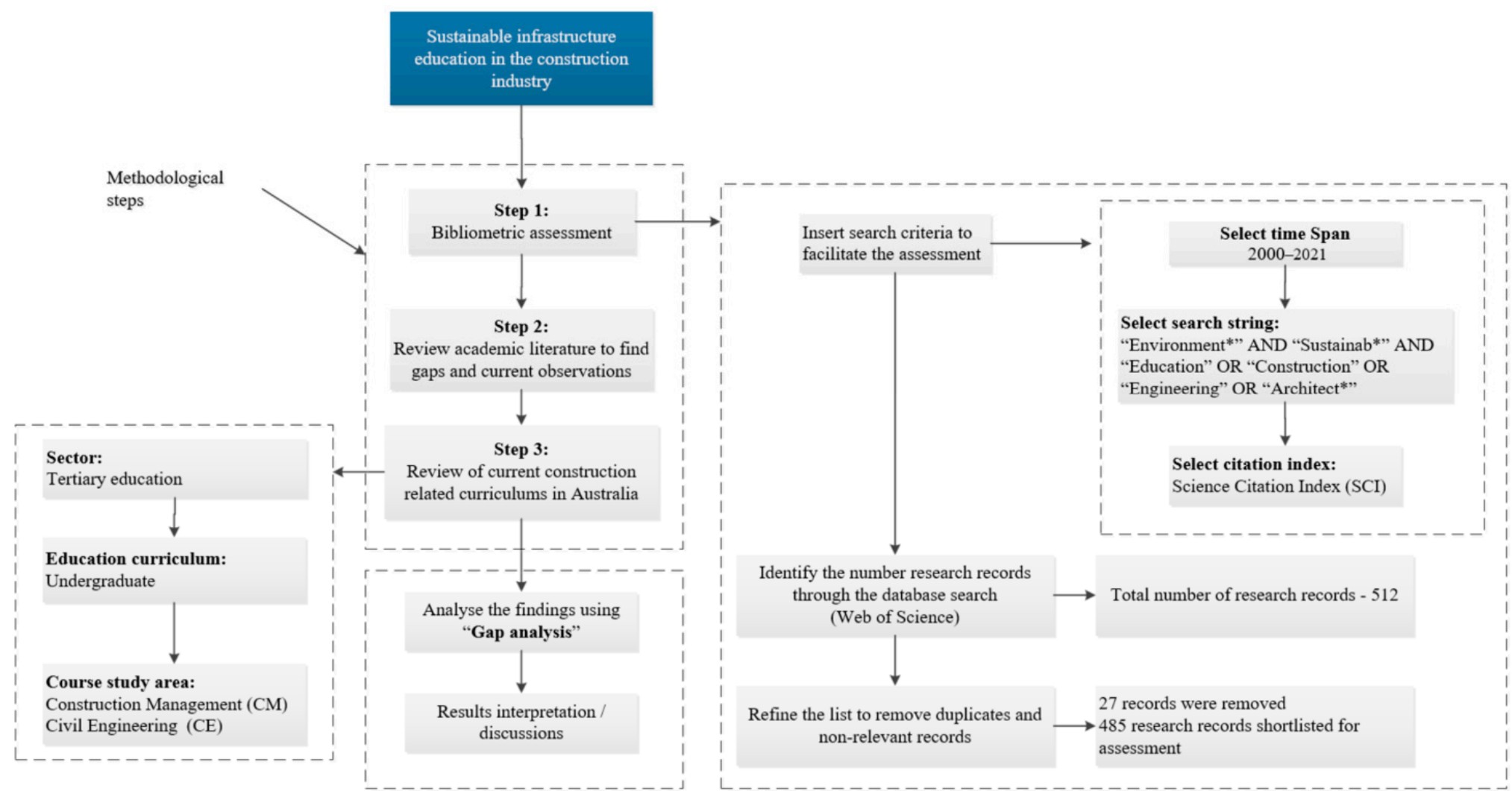

**Figure 1.** Research study scope and methodology.

### 3. Bibliometric Analysis Results

The timeline of the published articles in environmental sustainability in construction and related industries are highlighted in Figure 2. Based on the observations, the publication frequencies of articles can be discussed in three major time splices with a seven-year duration for each time splice. The initiation time splice from 2001–2007 had a constant rate of publications with less than ten publications per annum. The research interest initiation in the early 2000s can be linked to the adoption of the Kyoto Protocol (KP) on 11 December 1997 [16]. Since the adoption of the KP, research studies have focused on the introduction of education for environmental sustainability. The second time duration from 2007 to 2014 is presented as the development time splice due to an increased research focus on environmental sustainability in the construction industry. The initiation of the KP future targets and emission reduction strategies contributed to a gradual increase of research emphasis with an average annual publication of around 15 to 25 articles. The prominent time splice from 2014–present signifies the highest number and highest rate of publications with a sharp increase of publications in each year. This can be related to the first commitment period, from 2008–2012, of the KP emission reduction strategies (5% reduction as compared to the 1990 levels). With the recognition of impediments, focuses, and future directions, extensive research efforts were concentrated on environmental sustainability in construction. Moreover, the introduction of the UN SDGs in 2015 significantly influenced the research focus, and the number of publications almost doubled from 2016 to 2017. These results exemplify that research and education uptake are often connected with worldwide and nationwide initiatives and policy introduction.

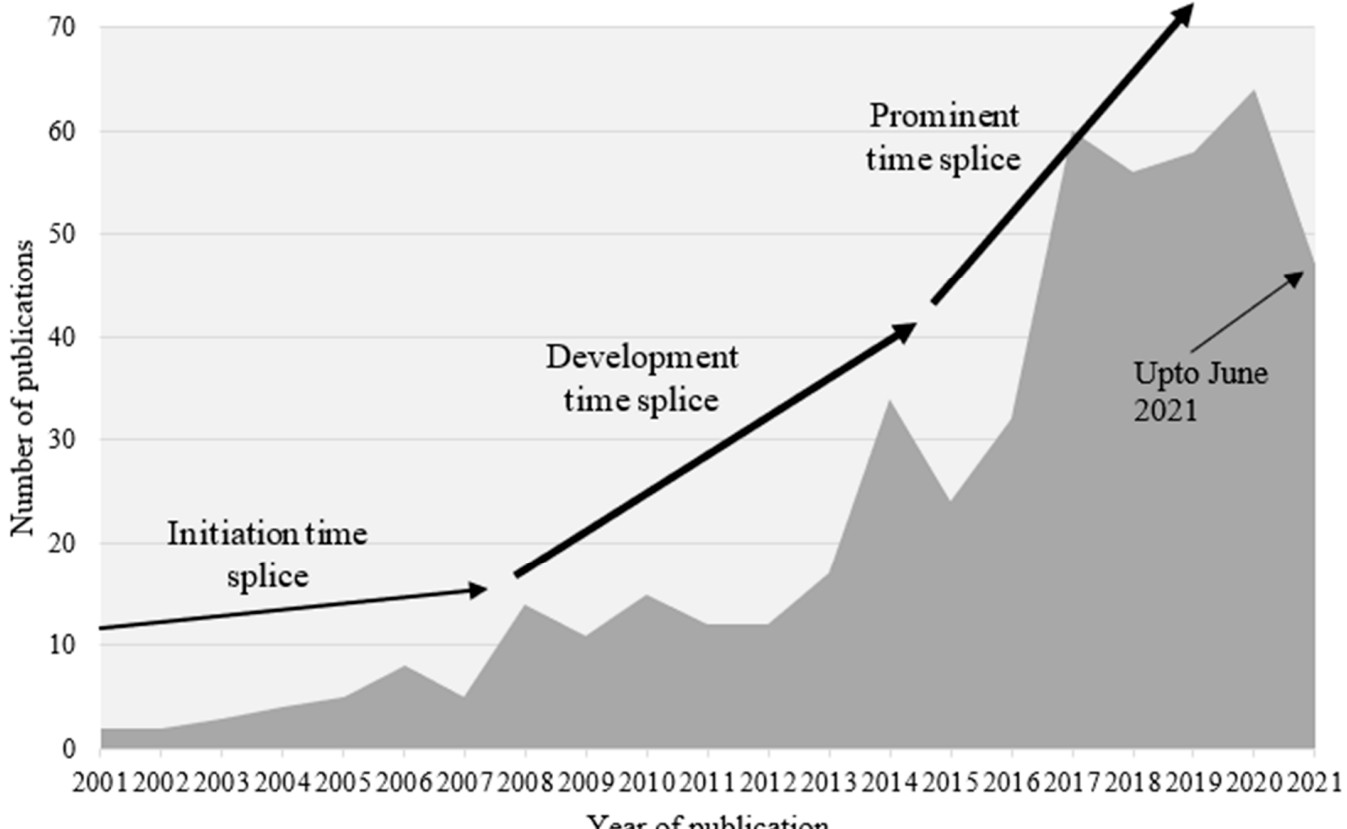

**Figure 2.** Timeline of related published articles.

The top four journals with their article production numbers are illustrated in Figure 3. As highlighted, since 2016, the top five journal publications have significantly increased the publication rate of research on sustainability education in the infrastructure construction industry. The publication rate for the journals with a wider scope of publication, such as *Sustainability* and *Journal of Cleaner Production*, is considerably high as compared to journals with a specific focus. This may be due to the two journals having "sustainability" outlined in the aims and scope. Significant high publication numbers in the *Sustainability* journal may indicate the authors' desire to publish in open-access journals with the intention of reaching a wider readership. The *Journal of Professional Issues in Engineering Education and Practice* was replaced by the *Journal of Civil Engineering Education*, and the current scope includes teaching methods, professional obligations, and the principles of formal education in civil engineering. These findings signify that the interest in implementing sustainability teaching strategies and techniques in construction and civil engineering education is significantly increasing.

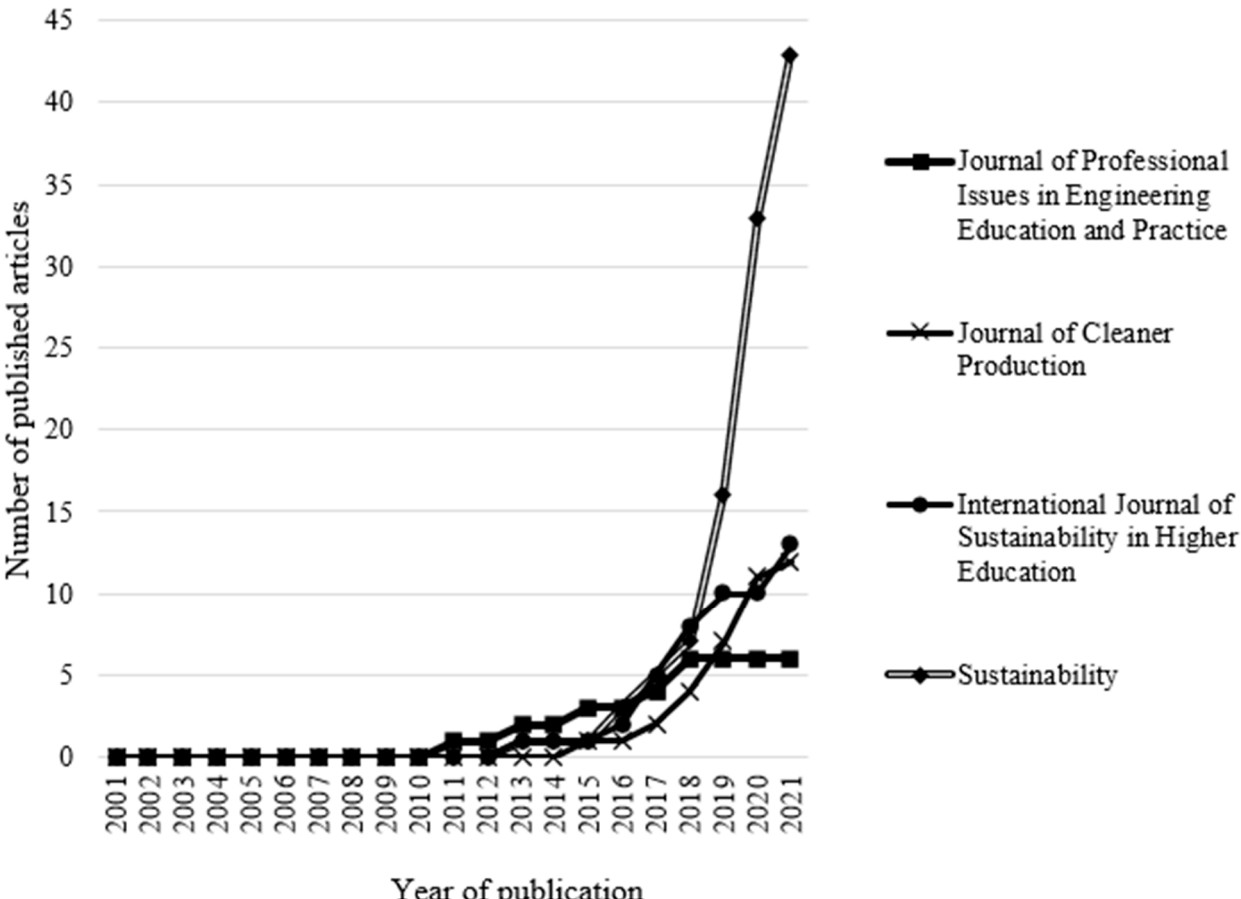

**Figure 3.** Top journals with related article production.

The key themes' evolution and the keywords' trending are highlighted in Figures 4 and 5. The size of each circle in Figure 4. indicates the significance of each research topic considered in the analysis. The circles in Figure 5. at a given year reflect that the corresponding keyword was used more in that year, and the size of the circle reflects how frequently the keyword was used. The bigger the circle size, the more that keyword has been used in scientific publications. These findings could facilitate the understanding of current research trends and future focuses in the educational and research fields related to sustainability education. Figure 4 indicates that the development of sustainability frameworks to address infrastructure and community aspects is a recently developed specialized research discipline. However, the focus has diverted towards research areas such as travel behaviour and sustainable transportation modes, while

education for sustainability has reduced its importance. Education and HE themes related to sustainability are at the basic level or travelling with less interest. Health- and energy-related topics in the infrastructure construction sector have also gained significant research interest over the past decade. The trending terms graph in Figure 5 reveals that "buildings" as an infrastructure type had more focus on environmental sustainability education, while other infrastructure types were seldom mentioned. However, the terms "cities" and "eco system" suggest there is a growing interest in sustainability education at the urban level as compared to the project level. Keyword terms such as "cars" and "health" further signify the considerable research attention towards sustainable behaviour and the use of infrastructure assets such as roads.

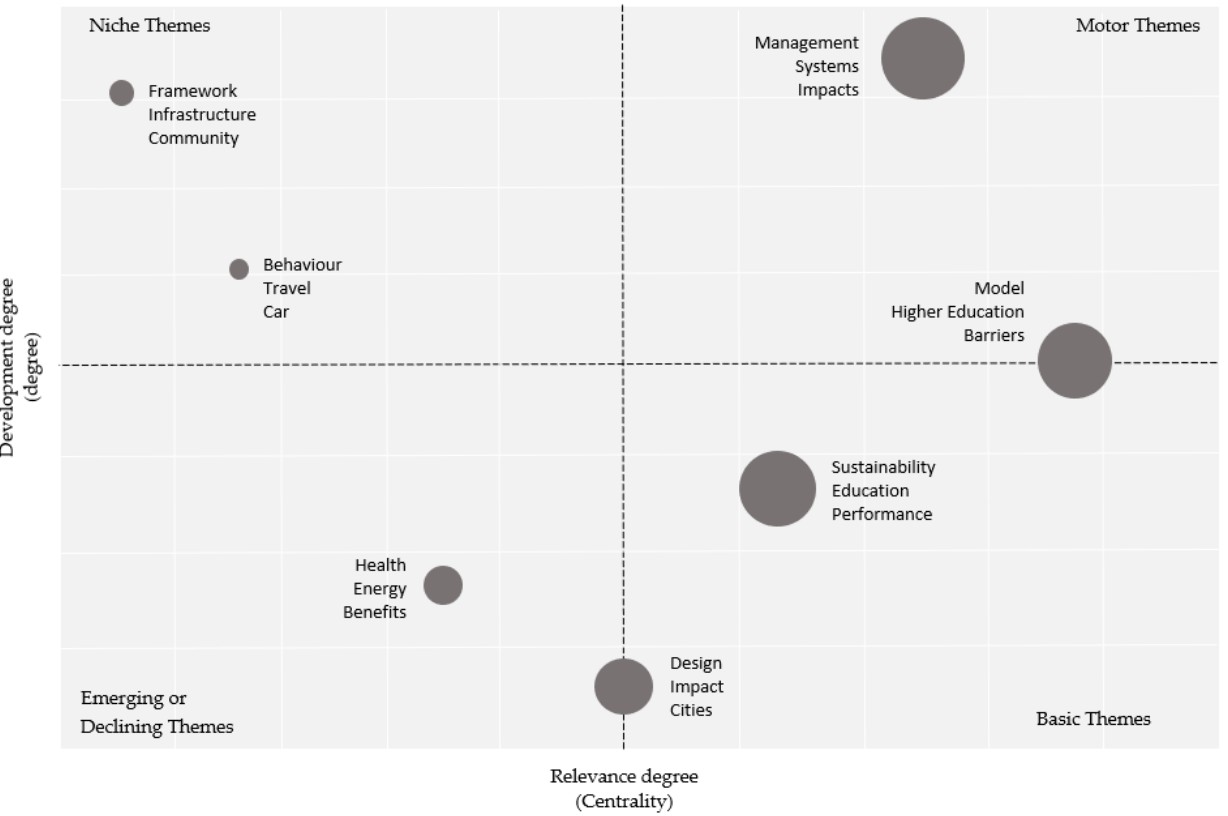

**Figure 4.** Key research themes' evolution.

A "three plot" between authors vs. countries vs. keywords, as shown in Figure 6, was derived to gain an understanding of the level of sustainability education in different countries across the globe. The United States of America (USA) and the United Kingdom (U.K.) are the leading countries to implement sustainability and sustainable development in their engineering HE curriculum. In addition to these two countries, China and Australia have made significant efforts to integrate sustainability into the Higher Education (HE) curriculum. Despite showing a keen interest in sustainability, Australia has not focused significantly on green infrastructure construction and environmental sustainability in the HE engineering curriculum over the past few decades. The term green infrastructure often refers to a facility or an asset that is designed and constructed to improve the environmental performance and economic benefits and enhances social life. Specific knowledge of design and construction concepts is required to effectively construct a green infrastructure asset. The results also indicate that sustainability education is divided between "engineering education" and "environmental education". This suggests that environmental sustainability concepts are either embedded in engineering curricula or stand alone as an environmental course. The annotations in Figure 6 further illustrate that education and knowledge development related to the policy and governance aspects of sustainable and green infrastructure

have seen decent advancements. However, another main observation is that "e-learning" and technology-related sustainability education has had a weak development and needs future improvements and research focus to identify technology integration possibilities. Moreover, a combined interpretation between the authors, countries, and keywords indicates that several studies conducted in different countries such as Canada, India, France, and Italy have published only a handful of studies on green/sustainable infrastructure education and, thereby, have no links to top publishing authors.

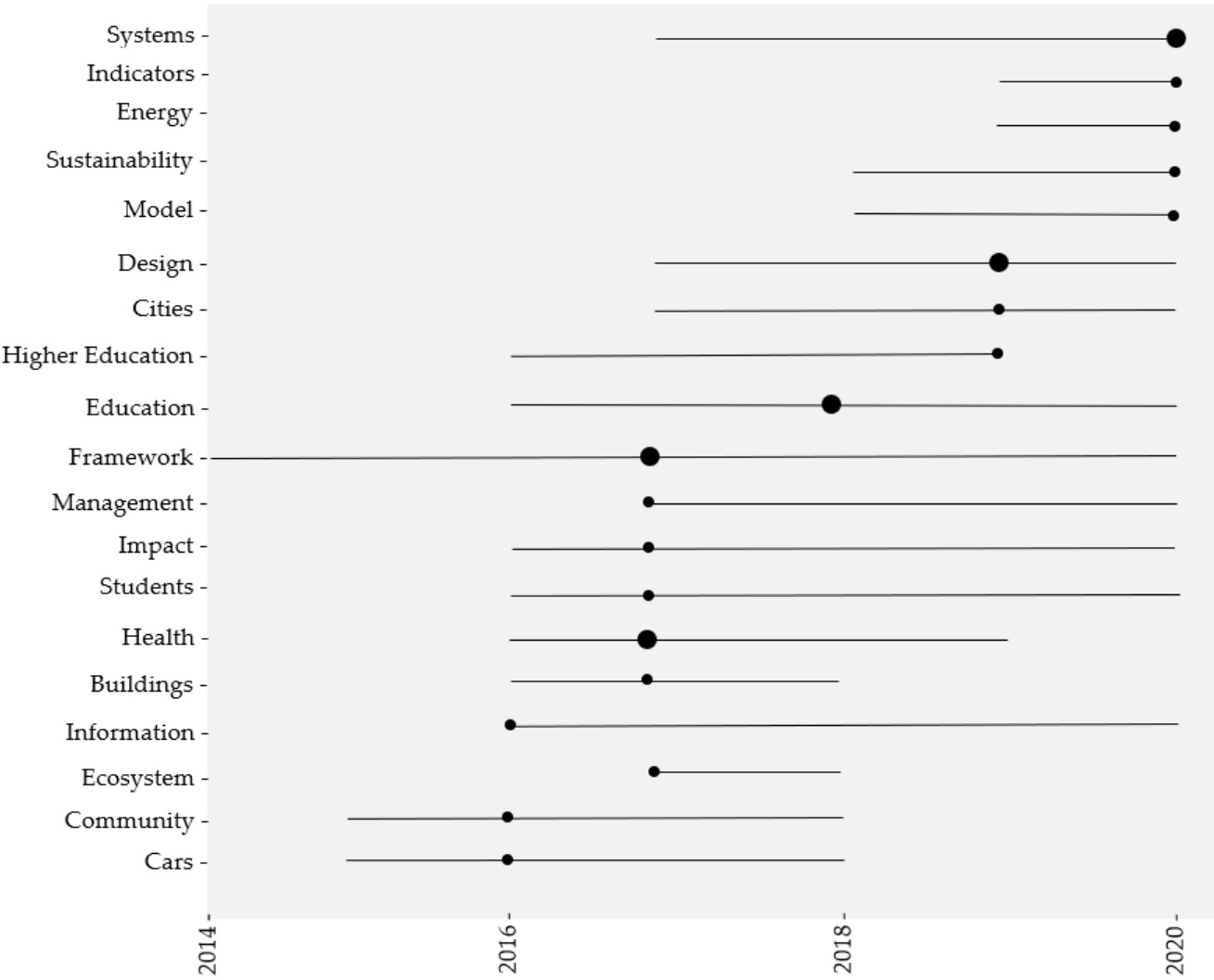

**Figure 5.** Trending keyword terms related to sustainable education in the construction industry.

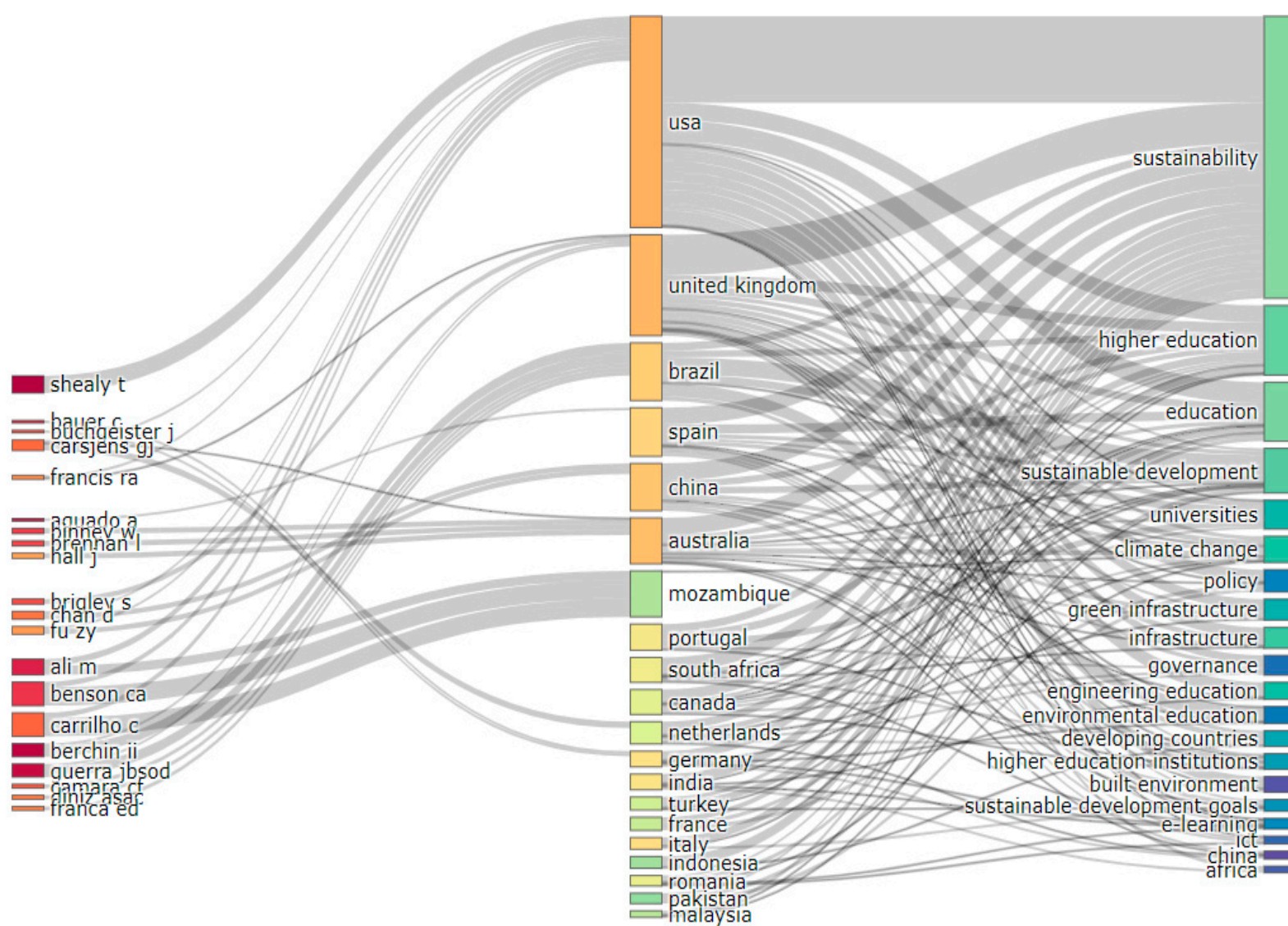

**Figure 6.** Three plot: authors vs. countries vs. keywords.

## 4. Findings from Previous Studies

### 4.1. Barriers to Sustainable Construction Practices

Despite the urgency, the uptake and development of environmentally sustainable practices in Australia's building and construction industry have failed to meet expectations. A wide range of barriers has hindered the transition to sustainable construction practices in Australia [17]. These barriers can be classified into three groups: resource issues, institutional, and psychosocial [10]. Financial constraints are one of the major resource barriers to implementing environmentally sustainable practices in construction [11,18,19]. Cost is still the governing factor when selecting materials [20]. There are often increased direct upfront costs with the procurement of sustainable materials, and at least long-term time and money savings need to be justified to incentivise contractors. Life cycle costing analyses are thus required to confirm cost savings over time [18]. This presents an additional barrier to industry due to possible human resource constraints and expertise limitations [20]. A lack of standards and data on material performance also hinders the use of sustainable materials, particularly recycled materials, as concerns exist with consistency and quality [21,22]. Table 1 exemplifies the sustainability focus of previously published literature on building and construction education.

As buildings emit 25% of Australia's total greenhouse gas emissions, the energy performance of buildings has attracted significant research attention. Martek et al. [17] investigated the technical and social interactions hindering the transition to a sustainable building industry in Australia by conducting focus group interviews with sustainability experts and practitioners. Whilst confirming known technical impediments, various social factors were uncovered as a deeper barrier to a true sustainable transition. This included vested interests and confused and unconvinced end-users. Hurlimann et al. [10] also found that a lack of client demand and sustainability awareness were plaguing the Australian building industry. Sustainability rating tools have been both praised and criticised for their role in improving sustainable outcomes. Rating schemes for buildings, such as GreenStar and NABERS, have been critiqued for not producing tangible outcomes and not holistically considering the whole life cycle of the building from planning through to deconstruction and the impact on the surrounding built environment [17]. The limitations of the regulatory framework in Australia have been highlighted in the literature as a barrier to sustainable practice. The existing standards are complex, slow to update, and feature limited accessibility [10]. Additional institutional barriers include a lack of leadership across the industry, a passive government with no long-term vision, and various requirements across states and territories [17]. Technical barriers include a lack of skilled contractors and subcontractors without a sufficient understanding of energy-efficient design principles [23,24]. Furthermore, barriers to implementing sustainable construction practices are filtering down to inhibit education in this space. A lack of demand for sustainable construction reduces funding and opportunities for research projects, which in turn restricts the development of a technical base for educational program development [25].

### 4.2. Sustainability in Built Environment Courses

The inadequate capabilities of graduates and non-existent professional development programs are also hindering progression in the construction industry in relation to sustainability [10]. Dhakal and Chevalier [11] argued that skilled personnel and expertise availability are necessary for Green Infrastructure (GI) construction and suggested that university education, research, and training play essential roles in sustainable development. Oyedele et al. [21] also recommended further research and training to improve the uptake of recycled materials in construction projects and stated that academic institutions are the key, due to the strong and well-recognised link between construction education and the sustainability competencies of early-career professionals. Ajayi et al. [20] corroborated this sentiment by stating that improved sustainability education, training, and awareness of construction professionals are required. However, the blame cannot be solely placed on training and education providers. Construction companies are required to play their

part by investing in education, training, and professional development to upskill their workforce [26] and to support research and development [25]. Strong relationships between industry and academia are critical in addressing this disconnection between education providers and industry needs [9]. Construction companies may rely on the competencies of newly appointed graduates to improve sustainability practices. Anh [27] surveyed 87 construction companies in the USA on their green building capabilities and experiences and found that 65% of companies surveyed expect university graduates to possess green building skills and knowledge including fundamental concepts, green rating tools, life cycle costing, and the green building design process. Sayce [9] reported this value to be 47% in the U.K., with a further 22% of organisations planning to include sustainability skills requirements in the next round of recruitment. Despite this, traditional and transferrable skills remained the priority over sustainability competencies during recruitment. This may be dissuading students from pursuing sustainability units and projects whilst at university and, instead, prioritizing conventional technical knowledge.

Sustainability education in construction at the HE level can include different categories and levels of competency development based on the curriculum design [7]. In Australia, the competency development at different levels are defined through learning outcomes and is facilitated through the Australian Qualifications Framework (AQF). The AQF provides a national set of criteria that must be met by HE institutions for all qualifications in Australia. The framework is centred on the AQF levels, which are descriptive criteria relating to the relative complexity and/or depth of achievement for formal qualifications. The AQF levels range from 1–10 and increase with complexity. Typically, the first year of a Bachelor's program in Australia aligns with AQF 5 and increases by one level every year of study. Lim et al. [7] categorised sustainability education in construction into background knowledge and concepts, policies and regulations, environmental, social, economic, technology, and innovation. The study also used the students' perception survey to capture sustainability understanding and knowledge. While the categorisation is expressive, the depth of learning in each subject was not comprehensively analysed. Kevern et al. conducted a case study analysis to investigate the inclusion of sustainability education for civil engineers on green buildings and sustainable infrastructure [28]. The framework proposed the use of a commercially available green rating tool, i.e., Leadership in Energy and Environmental Design (LEED), to design the course activities and classwork. In addition, the framework suggested improvements of ten major concepts of sustainability by using fourteen weeks of course curriculum. However, there is uncertainty over the adoptability of a building-focused rating tool into sustainable infrastructure projects due to the salient differences of sustainability requirements for building and infrastructure projects. A study conducted by the Queensland University of Technology (QUT) explored the sustainability inclusion in the construction management curricula [7]. The results revealed that sustainability integration is rather a horizontal approach covering sustainability concepts only in its general units. The units in the course covered general sustainability concepts to more in-depth considerations such as policies, technologies, and innovations. The topics covered in the sustainability-related units revealed that the content is focused on buildings' and rating tools' assessment and legislation aspects are focused on infrastructure projects. Similarly, a study conducted in the USA proposed a framework for incorporating sustainable concepts into a civil engineering course [8]. The focus of the content was based on using the Leadership in Energy and Environmental Design (LEED) green rating system to introduce a green building. The proposed course schedule included several in-depth environmental sustainability assessment methods such as Life Cycle Assessment (LCA). However, the emphasis was only on buildings, and other infrastructure construction projects were not considered in the study.

An ideal solution would be to introduce a curriculum that can address the depth and breadth of environmental sustainability concepts. However, thorough incorporation of environmental sustainability into construction and engineering programs does present its own challenges. Sustainability is inherently multi-disciplinary and often indefinable. Hence, prescriptive and descriptive approaches are required, which renders traditional, discipline-based teaching methods ineffective when teaching sustainability. As a result, horizontal integration of sustainability concepts into core curricula becomes impractical as teachers typically prefer a single teaching style and are burdened by training to learn or adopt new ones [29]. Garud [25] found that most sustainability knowledge is imparted when descriptive and interactive methods of teaching are integrated with core subjects. Other researchers have also recommended the adoption of student-centric approaches to teach sustainability [30]. Moreover, the interest and the knowledge levels of the respective academic staff are also governing factors that contribute to the comprehensiveness of the sustainability content covered in units. Often, the sustainability content tends to be a general introduction only, if the staff does not possess the required teaching qualifications and experience.

### 4.3. Continuous Professional Development

Many previous studies (Table 1) have argued that sustainability education can be provided to early-career professionals by their employer through Continuous Professional Development (CPD) programs [9]. CPD programs provide practitioners with the opportunity to keep up to date with new knowledge and developments. Any type of learning, formal or informal, can be considered as CPD, including but not limited to conferences, seminars, formal education and qualifications, personal study, and in-house training. Despite being considered as a valuable approach, the value of current CPD activities in building and construction has been questioned. An interview study of architects conducted in Scotland described CPD activities as mostly marketing opportunities for new products, as opposed to genuine education [24]. Additional concerns raised by the authors included the inability to evaluate the effectiveness of CPD events. A similar view has been expressed regarding CPD programs for builders in Australia. Graham and Warren-Myers [31] investigated the efficacy of a sustainability education program for Australian building professionals in the residential sector by interviewing course participants. Overall, the specific program was viewed as a valuable educational opportunity; however, implementation rates of the course knowledge into business practices was low due to the pre-existing sustainability education of the participants and the introductory nature of the content. The interviewees in the study showed a desire to pursue sustainability education and communicated their need for longer-term, comprehensive, and practical education opportunities. This was mainly due to the limitations associated with the short, seminar-style approach adopted by most existing programs. Furthermore, professional level recognition of the CPD activities at the organizational level seems to be limited, which further limits the desire of graduates within the infrastructure construction sector.

**Table 1.** Summary of previous study findings.

| No. | Study Focus | Study No. | | | | | | | | | | | | | | |
|---|---|---|---|---|---|---|---|---|---|---|---|---|---|---|---|---|
| | | 1 | 2 | 3 | 4 | 5 | 6 | 7 | 8 | 9 | 10 | 11 | 12 | 13 | 14 | 15 |
| 1 | Sustainability education focused on buildings | √ | √ | √ | √ | | √ | √ | √ | √ | √ | | √ | √ | | √ |
| 2 | Sustainability education focused on built environment and other construction industries | | | | | √ | | | | | | √ | | | √ | |
| 3 | Social and ethical issues of sustainability education in construction | | | | | | | | √ | | | | | | | |
| 4 | Sustainability education framework for higher-education-level construction curriculum | √ | | | | | | √ | | √ | | | | | √ | |
| 5 | Effective teaching strategies to facilitate sustainability education | | | | | | | | | | | | √ | √ | | |
| 6 | Developmental changes to built environment curriculum | | | | | | | | | | | | √ | | √ | |
| 7 | Multi-disciplinary approach to facilitate sustainable engineering education | | | | | | | | | | | | | | √ | |
| 8 | Embedding sustainability within the construction curriculum | | √ | | | | √ | | | | | | | | | |
| 9 | Use of smart technologies to facilitate sustainability education | | | √ | | | | | | √ | √ | | | | | |
| 10 | Higher education students' interest in sustainability-related topics | √ | √ | | | √ | √ | | | | | | | | | √ |
| | Reference | [32] | [33] | [34] | [28] | [35] | [36] | [37] | [38] | [39] | [40] | [41] | [42] | [43] | [44] | [45] |

## 5. Review of Existing Course Curricula in Australia

### 5.1. Study Setup and Analysis Basis

The comparative review of existing course curriculum aimed at examining environmental sustainability concepts in Construction Management (CM) and Civil Engineering (CE) courses at Victoria University was undertaken as part of this study. The study also focused on capturing the infrastructure sustainability considerations within both curricula. Both the CE and CM programs are four years in duration with eight subjects (units) per year and 12 Credit Points (CP) for each subject (unit) corresponding to a total of 384 CPs (32 units). The CM course consists of 26 core units, 2 capstone units (research project), and 4 minor units distributed across 4 years. The course addresses four key major areas of knowledge including technical, legal, management, and economics. The CM course at Victoria University has received full accreditation from the Australian Institute of Building (AIB). The CE course is accredited by Engineers Australia (EA) and features 30 core units and 2 capstone units (research project).

Required environmental sustainability competencies are categorised into seven key environmental sustainability topics, as shown in Table 2. These topics were derived following the review of previously published literature [35,37,39,44] and sustainability rating tools [4,46]. The application and depth of environmental sustainability knowledge and skills in the CM and CE curriculum is integrated based on AQF Levels. Table 3 illustrates the CM and CE undergraduate programs at Victoria University with the corresponding years of study, AQF levels, and environmental sustainability subjects (units). The environmental sustainability area of focus of each unit was identified to obtain an understanding of the level of environmental sustainability consideration in curricula for each year of study. In addition, learning areas related to environmental sustainability were assessed for each unit based on the key verbs used in the relevant unit learning outcomes. The learning components were categorised into cognitive, communication, creative, and technical areas to understand the learning focus related to environmental sustainability in each unit of the course. Cognitive learning and education focus on improving the learning capacity including intelligence, social, and emotional aspects. "Creative" corresponds to improving imagination and critical thinking, which leads to managing risks, independent working, and flexibility. "Communication" corresponds to skills related to the exchange of facts, ideas, and opinions for the application of theories and principles. "Technical" corresponds to the theories, principles, and understandings to facilitate related subject learning. The corresponding learning verbs, for each learning area and belonging to the relevant AQF level, are defined in Table 4 [47].

**Table 2.** Key environmental sustainability education categories.

| SI No. | Environmental Sustainability Topic Category | References |
|--------|---------------------------------------------|-----------|
| SI 1 | Environmental sustainability practice and knowledge | [7,35] |
| SI 2 | Sustainable design concepts | [39] |
| SI 3 | Sustainable policies, standards, and regulations | [7,48] |
| SI 4 | Sustainable site and workforce management | [4,39] |
| SI 5 | Sustainable assessment | [8,41] |
| SI 6 | Sustainable procurement, technologies, and innovations | [4,39] |
| SI 7 | Sustainable materials | [7,39] |

**Table 3.** Victoria University's CM and CE programs: qualifications and subjects in each year of study.

| Year of Study | AQF Level | Units Related to Environmental Sustainability in CM and CE Courses |
|---|---|---|
| 1 | 5 | **CM Course**<br>■ NBC1111—Fundamentals of Building Construction<br>■ NBC1112—Building Science<br>■ NBD1100—Built Environment and Communication Skills<br>■ NBD1101—Building Design and Documentation<br><br>**CE Course**<br>■ NEF1103—Engineering and Community<br>■ NEF1104—Problem Solving for Engineers<br>■ NEF1204—Introduction to Engineering Design |
| 2 | 6 | **CM Course**<br>■ NBC2003—Building Systems and Services<br>■ NBC2004—Building and Construction Studies<br>■ NBC2005—Building Materials<br>■ NEA2201—Building Development and Compliance<br>■ NBC2101—Building and Construction Surveying<br><br>**CE Course**<br>■ NEC2103—Engineering Materials and Construction<br>■ NEC2204—Highway Engineering |
| 3 | 7 | **CM Course**<br>■ NBC3001—High-rise Development and Compliance<br>■ NBC3003—Building Services Management<br>■ NBC3006—Construction Site Operations<br>■ NBC3204—Complex Construction<br><br>**CE Course**<br>■ NEC3103—Hydrology and Water Resources |
| 4 | 8 | **CM Course**<br>■ NBC4002—Advanced Construction<br>■ NBC4003—Cost Planning and Control<br>■ NBC4101—Construction Management<br><br>**CE Course**<br>■ NEC4101—Environmental Engineering 1<br>■ NEC4172—Urban Development and Transportation<br>■ NEC4206—Advanced Engineering Design<br>■ NEC4207—Engineering Applications |

Learning outcomes, learning content, assessments, and evaluation criteria for each unit (across all 32 units) were analysed with the aim to assess the integration of the environmental sustainability competencies in the two curricula. The aim of the comparative analysis was to obtain a general understanding of the embedment of sustainability in the current CM and CE curricula at Victoria University, Australia. However, the findings cannot be generalized for sustainability inclusions to other institutions due to inconsistencies in teaching, delivery, and evaluation methods. The findings can still be used as a basic comparative tool to obtain a general understating of the environmental sustainability considerations for the CM and CE curricula. Besides, the study did not consider architecture, design, and other built-environment-related curricula and only focused on the CM and CE curricula.



**Table 4.** Matrix representation of learning outcome verbs corresponding to AQF levels and learning areas [47].

| Learning Areas | Level of Qualification (AQF Level) | | | |
| --- | --- | --- | --- | --- |
| | AQF 5 | AQF 6 | AQF 7 | AQF 8 |
| Cognitive | adapt, analyse, assess, attribute, budget, calculate, catalogue, categorise, classify, compare, contrast, coordinate, determine, diagnose, discuss, elaborate, evaluate, examine, extrapolate, formulate, integrate, interpret, investigate, locate, modify, organise, paraphrase, prioritise, quantify, reconstruct, relate, retrieve, review, role-play, solve, substantiate, summarise, synthesise, tabulate, troubleshoot, verify | adapt, adjudicate, analyse, annotate, appraise, arbitrate, argue, assess, attribute, authenticate, calculate, challenge, compare, conceptualise, conclude, contextualise, contrast, critique, debrief, decode, deduce, defend, deliberate, derive, determine, diagnose, discriminate, engineer, evaluate, exemplify, extrapolate, formulate, infer, integrate, interpret, investigate, judge, justify, map, mediate, modify, optimise, prescribe, probe, propose, prove, qualify, quantify, recommend, reconstruct, reflect, relate, resolve, review, scrutinise, solve, substantiate, survey, troubleshoot, test, verify | analyse, annotate, appraise, arbitrate, argue, assess, authenticate, challenge, commentate, conclude, corroborate, critique, critically review, critically reflect, conceptually map, contextualise, cross-examine, decode, debrief, deduce, defend, deliberate, derive, discriminate, diagnose, dispute, distil, extrapolate, forecast, hypothesise, infer, interpret, inquire, interrogate, investigate, justify, mediate, predict, prescribe, probe, propose, prove, qualify, quantify, rationalise, recommend, reconstruct, reflect, resolve, substantiate, survey, validate | analyse, arbitrate, argue, authenticate, commentate, critically review, critique, cross-examine, conceptually map, corroborate, deconstruct, deduce, derive, dispute, explicate, infer, interpret, interrogate, justify, posit, postulate, propose, qualify, rationalise, recommend, resolve, reverse-engineer, theorise, triangulate, validate |
| Communication | advise, articulate, clarify, collaborate, discuss, exemplify, explain, guide, introduce, manage, orient, present, propose, question, re-enact, report, script, translate | advise, argue, articulate, analogise, collaborate, construe, consult, convince, co-operate, coordinate, debate, discourse, elaborate, elicit, exemplify, exhibit, negotiate, network, persuade, present, question, report, role-play, summarise, translate | advocate, adjudicate, allegorise, argue, construe, consult, conciliate, convince, debate, discourse, elicit, elucidate, exemplify, extrapolate, negotiate, network, persuade, report, role-play | advocate, argue, canvass, conclude, convince, debate, discourse, distil, elucidate, exemplify, interview, persuade, present |
| Creative | analogise, brainstorm, choreograph, compose, depict, design, devise, dramatize, engineer, exhibit, fabricate, fashion, illustrate, imagine, initiate, plan, optimise, sketch, storyboard | adapt, allegorise, brainstorm, compose, choreograph, design, devise, exhibit, fabricate, fashion, initiate, map, modify, plan, role-play, scope, strategize, storyboard, transform | adapt, brainstorm, choreograph, compose, devise, initiate, role-play, scope, strategize, transform | compose, devise, hypothesise, innovate |

**Table 4.** *Cont.*

| Learning Areas | Level of Qualification (AQF Level) | | | |
| --- | --- | --- | --- | --- |
| | **AQF 5** | **AQF 6** | **AQF 7** | **AQF 8** |
| Technical | adjust, administer, apply, assemble, build, budget, cite, craft, create, demonstrate, develop, differentiate, distinguish, embellish, employ, estimate, experiment, extend, footnote, generate, graph, inspect, journal, locate, manipulate, measure, monitor, observe, operate, plot, practise, prepare, record, repair, revise, schedule, sequence | apply, budget, calculate, catalogue, cite, clarify, classify, compare, compute, create, demonstrate, develop, differentiate, distinguish, discuss, dramatize, elaborate, embellish, employ, estimate, experiment, explain, footnote, graph, inspect, locate, manipulate, manage, monitor, illustrate, observe, operate, organise, paraphrase, plot, prioritise, retrieve, revise, schedule, sketch, tabulate | adapt, advise, analyse, analogise, articulate, attribute, budget, brainstorm, clarify, collaborate, compute, compose, conceptualise, construct, construe, consult, coordinate, design, determine, devise, dramatize, engineer, evaluate, exhibit, experiment, fabricate, formulate, initiate, integrate, judge, manage, modify, negotiate, monitor, modify, plan, present, prioritise, query, question, review, scrutinise, storyboard, substantiate, synthesise, tabulate, test, translate, troubleshoot, verify | adapt, allegorise, annotate, appraise, attribute, challenge, collaborate, compute, contextualise, defend, deliberate, design, diagnose, discriminate, elicit, estimate, evaluate, exhibit, experiment, extrapolate, forecast, formulate, judge, modify, monitor, implement, infer, inquire, interview, investigate, plan, predict, present, probe, prove, qualify, quantify, query, question, reconstruct, re-model, scope, solve |

*5.2. Curriculum Comparison of Environmental Sustainability Consideration Findings*

The investigations of contents, learning outcomes, and assessments in both the CM and CE curricula revealed that environmental sustainability was mainly embedded horizontally across units with a focus on achieving desired knowledge and skill outcomes.

Figure 7 illustrates the distribution of sustainability- and non-sustainability-related units in both CM and CE courses across the four years of study. The distribution considered both horizontal and vertical integration of sustainability concepts in unit curricula. Sustainability topics and considerations in CM units spanned uniformly across all four years, while first and fourth year units in the CE course incorporated the majority of sustainability-related contents. This is mainly because the CE course incorporated fundamental sustainability concepts in the first year and design-specific environmental sustainability aspects in the fourth year. On the contrary, the CM course covered a wide spectrum of sustainability including fundamental concepts, technical criteria, policies and legislation requirements, and innovations related to construction. Similar to the CE course, the fourth year units in the CM course focused on specialised environmental sustainability competencies related to design and management in construction.

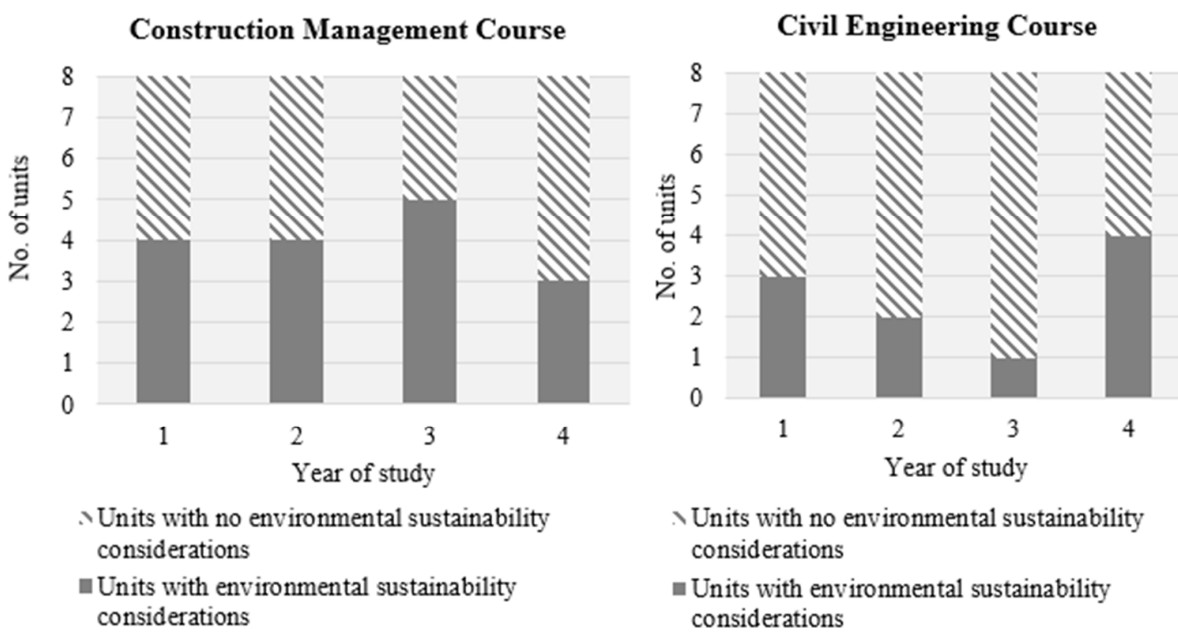

**Figure 7.** Distribution of environmental sustainability units by year of study.

To understand the focus and learning depth of key environmentally sustainable education areas, each CM and CE unit at different AQF levels were mapped as shown in Figure 8. Sustainability consideration in each unit was observed from learning contents and learning outcomes, whereas the depth of learning was measured through the AQF level considerations of each unit. The findings illustrate that sustainable design concepts, policies and standards, and sustainable procurement concepts are covered within both CM and CE courses across different AQF levels (Table 5). This exemplifies the high consideration of fundamental learning components in university curricula. Sustainability assessment, materials, and workforce management are the least considered aspects within the CM course. Some of these aspects, such as assessment of sustainability levels and the use of sustainable materials are considered at higher AQF levels, with a focus on independent decision-making on problem-based learning. In contrast to CM, the CE course is more focused on the design concepts of environmental sustainability facilitating engineering design, and sustainable assessment of different civil engineering applications including environmental, water, and materials engineering.

Figure 8 depicts the distribution of learning outcomes relating to sustainability and the corresponding learning areas for the CM and CE courses at Victoria University across AQF Levels 5–8. For each AQF level, the units containing sustainability-related learning outcomes are listed with the number of learning outcomes for each learning area highlighted. Each learning area is represented through the corresponding key verbs used in the learning outcomes of each unit, as per Table 4. The results indicate that units across all four years in both courses focus on improving cognitive skills related to environmental sustainability. In CM, the learning outcomes of AQF Level 8 units were more focused on the creative learning area, while AQF Level 5 was more focused on the cognitive and communication learning areas related to environmental sustainability. The technical learning areas in environmental sustainability were more embraced at AQF Levels 6 and 7. In contrast, the majority of the CE units focused, at all AQF levels, on improving cognitive and technical skills related to environmental sustainability. Thus, the CE students learn mainly the technical and fundamental environmental sustainability aspects as required to facilitate engineering designs and applications. These findings imply that both curricula cover cognitive skills related to environmental sustainability and a lack the application component. Thus, graduates and industry professionals require additional time and commitment to understand the practical implementation, including sustainable assessment and evaluation. There is a strong need for a structured and well-complimented program that could facilitate the sequential learning and required practical environmental sustainability in construction competencies' development.

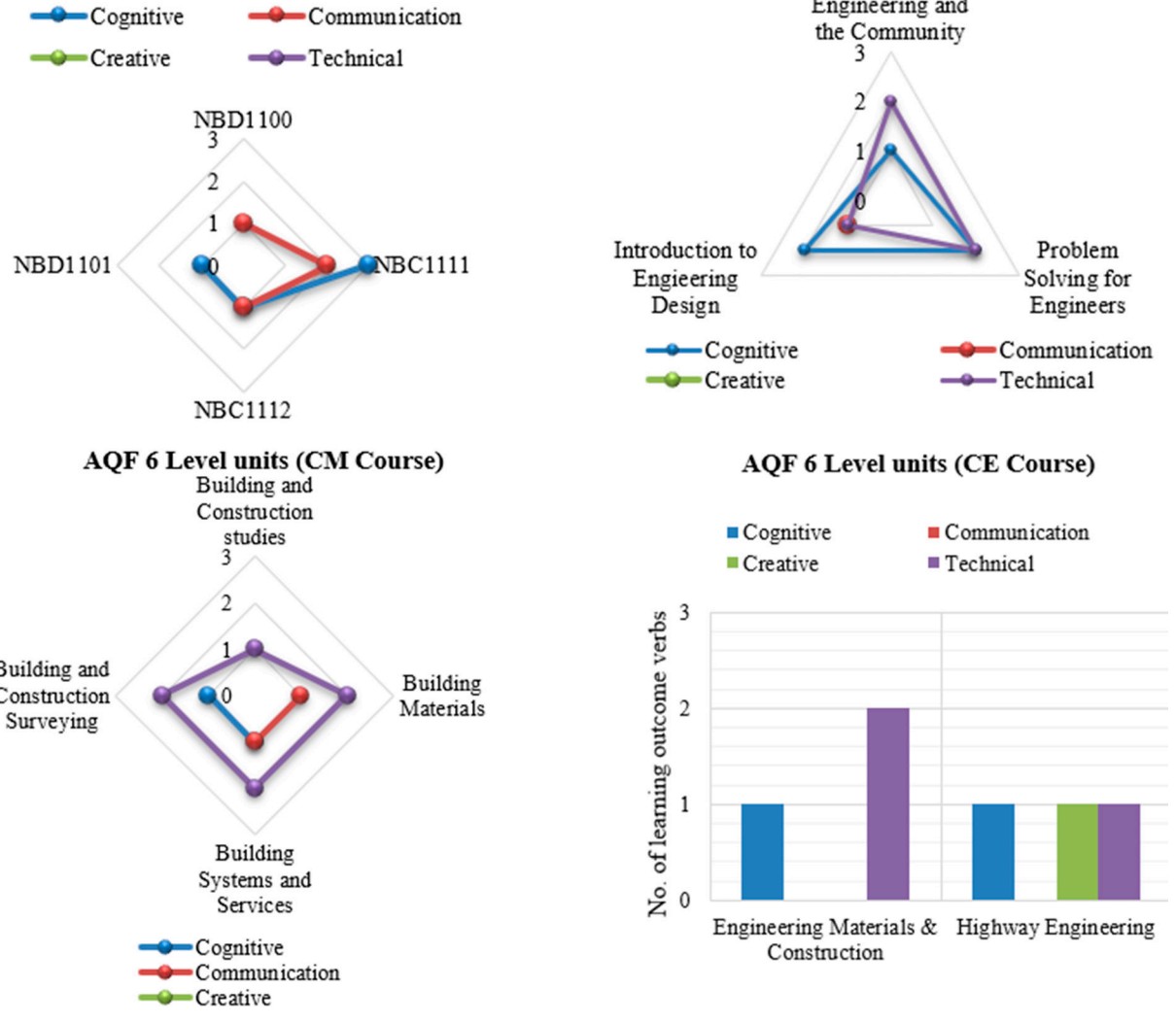

**Figure 8.** *Cont*.

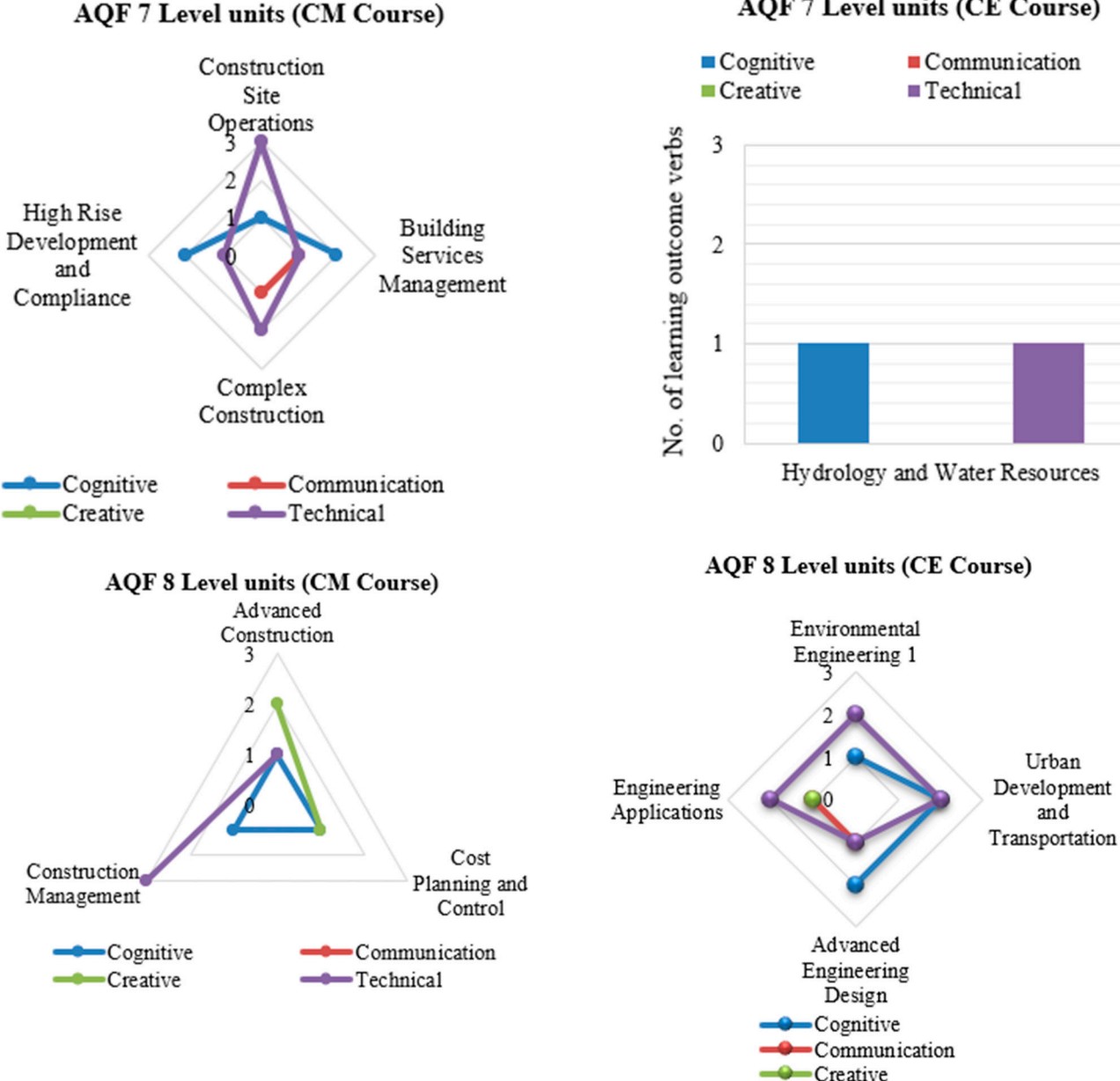

**Figure 8.** CM units across different AQF levels and AQF verbs' representation for the four learning areas.

**Table 5.** Environmental sustainability aspects in each unit of CM and CE courses at different AQF levels.

| Construction Management (CM) Course | AQF—Australian Qualification Framework | | | | | | | | | | | | | | | | |
| --- | --- | --- | --- | --- | --- | --- | --- | --- | --- | --- | --- | --- | --- | --- | --- | --- | --- |
| | AQF5 | | | AQF6 | | | | | AQF7 | | | | AQF8 | | | | |
| Category/Unit Name | NBC1111 | NBC1112 | NBD1101 | NBD1100 | NBC2003 | NBC2004 | NBC2005 | NEA2201 | NBC2101 | NBC3001 | NBC3006 | NBC3204 | NBC3003 | NBC4002 | NBC4003 | NBC4101 | Total |
| Environmental sustainability knowledge and concepts | | | | √ | √ | | | | | | | | √ | √ | | √ | 5 |
| Sustainable design concepts | | √ | √ | √ | √ | | | √ | √ | | | | √ | | √ | | 8 |
| Policies, standards, and regulations | | √ | | | √ | | | √ | √ | √ | | | √ | | | √ | 7 |
| Sustainable site and workforce management | | | | | | √ | | | | √ | √ | | | | | | 3 |
| Sustainable procurement, technologies, and innovations | √ | | | | | | | | | | √ | √ | √ | √ | | √ | 6 |
| Sustainable assessment | | | | | | | √ | | | | | | √ | √ | √ | | 4 |
| Sustainable materials | √ | | | | | | √ | | | | | √ | | √ | | | 4 |
| Total | 2 | 2 | 1 | 2 | 3 | 1 | 2 | 2 | 2 | 2 | 2 | 2 | 5 | 4 | 2 | 3 | |

| Civil Engineering (CE) Course | AQF—Australian Qualification Framework | | | | | | | | | | |
| --- | --- | --- | --- | --- | --- | --- | --- | --- | --- | --- | --- |
| | AQF5 | | | AQF6 | | AQF7 | | | AQF8 | | |
| Category/Unit Name | NEF1103 | NEF1104 | NEF1204 | NEC2013 | NEC2204 | NEC3103 | NEC4101 | NEC4172 | NEC4206 | NEC4207 | Total |
| Environmental sustainability knowledge and concepts | | √ | √ | | √ | √ | √ | | √ | √ | 7 |
| Sustainable design concepts | | √ | √ | | √ | | √ | √ | √ | √ | 7 |
| Policies, standards, and regulations | √ | | | | | | | | | | 1 |
| Sustainable site and workforce management | | | | √ | | | | √ | | | 2 |
| Sustainable procurement, technologies, and innovations | | | | | √ | | | | √ | | 2 |
| Sustainable assessment | | √ | | | | | √ | √ | √ | | 4 |
| Sustainable materials | | | | √ | | | | | | | 1 |
| Total | 1 | 3 | 2 | 2 | 4 | 1 | 3 | 3 | 3 | 3 | |

NBC1111—Fundamentals of Building Construction, NBC1112—Building Science, NBD1100—Built Environment and Communication Skills, NBD1101—Building Design and Documentation, NBC2003—Building Systems and Services, NBC2004—Building and Construction Studies, NBC2005—Building Materials, NEA2201—Building Development and Compliance, NBC2101—Building and Construction Surveying NBC3001—High-rise Development and Compliance, NBC3003—Building Services Management, NBC3006—Construction Site Operations, NBC3204—Complex Construction NBC4002—Advanced Construction, NBC4003—Cost Planning and Control, NBC4101—Construction Management. NEF1103—Engineering and Community, NEF1104—Problem Solving for Engineers, NEF1204—Introduction to Engineering Design, NEC2103—Engineering Materials and Construction, NEC2204—Highway Engineering, NEC3103—Hydrology and Water Resources, NEC4101—Environmental Engineering 1, NEC4172—Urban Development and Transportation, NEC4206—Advanced Engineering Design, NEC4207—Engineering Applications.

**6. Findings and Discussions**

The results from the bibliographic analysis, literature review, and course assessment can be further discussed based on four main categories: current advancements and future initiatives, barriers and impediments to successful implementation, requirements of an integrated approach towards CPD, and curriculum development and implementation. These findings will provide a better understanding of the current advances, impediments, and future research areas.

*6.1. Current Advancements and Future Initiatives*

The review of both previous research studies and two Victoria University curricula revealed that environmental sustainability capabilities and knowledge were focused heavily on the building industry, and limited emphasis was allocated to the infrastructure sector. This could be due to the high volume of building construction and related works. However, due to the rapid infrastructure development across many countries in the world, there is a contemporary requirement to train professionals in the infrastructure sustainability domain. However, infrastructure construction is often hard to capture due to the uniqueness of construction activities and the dynamic nature of the construction process. Therefore, the development of structured and effective training to facilitate systematic learning is crucial for expanding the infrastructure construction environmental sustainability professionals' databases. Social value creation is another critical emerging trend in the construction industry, and organizations are making valiant efforts to achieve these targets. However, often, it is extremely difficult to obtain sustainable social procurement in the infrastructure construction sector that could cater to the long-term benefits for both societies and organizations. Therefore, short courses on environmental sustainability on infrastructure construction targeting individuals with a low socioeconomic status could bridge both gaps with a single holistic solution. This would also provide an easy, yet promising career upskilling pathway for Technical and Further Education (TAFE), qualified construction technicians. However, the target content and qualification level need to be carefully tackled to reap the maximum benefits of such a course.

With several worldwide initiatives, countries are implementing policies and legislation to successfully integrate environmental sustainability with construction projects. Recent research studies have shown interest in effective policy development, governance, and legal aspects related to mandating environmental sustainability education for construction and engineering graduates. These initiatives would lead to students at all levels obtaining sustainability competencies in addition to the core technical competencies. Moreover, additional resources and facilities are required to deliver environmental sustainability education in infrastructure construction. This could be in the forms of acquiring additional software and active subscription for technical documentation for various rating systems, which require additional funding. Therefore, it is important for governments and the responsible organizations to allocate additional funding to facilitate the effective delivery of sustainability education and potentially reap long-term returns on investment with many sustainability benefits.

*6.2. Barriers and Impediments to Successful Implementation*

A lack of motivation is one of the major psychosocial barriers that hinders the uptake of environmental-sustainability-related expertise. The infrastructure construction sector is often busy, and the demand for technical experts alone is quite high, especially in the case of Australia. With limited additional time and high pay-scales, graduates are often less interested in investing to gain additional expertise. Moreover, the majority of the civil engineering and construction management graduates have given preference to project management post-graduate expertise with the prime aim of travelling through the promotion ranks in their organization [49]. Limited recognition for environmental sustainability expertise at the infrastructure construction organization level is also another reason leading to a lack of motivation. However, with growing interest in the sustainability

domain, organizations are starting to realize the importance of environmental sustainability, with many introducing their own corporate sustainability targets. Therefore, a structured process is required to recognise the qualifications and professional development in the environmental sustainability area. This is a decisive factor in acknowledging the commitment towards developing environmental sustainability expertise. The extent to which Australian organisations value sustainability knowledge in built environment graduates is not well understood and requires further research.

Complexities in teaching and delivery comprise another barrier to effectively implementing education on sustainable infrastructure. Often, the construction techniques and processes are unique and change rapidly according to technology advancements. In addition, acquiring industry involvement consistently is a challenge, which results in concerns about dynamic learning experience. Despite that the introduction of technology-integrated teaching could eliminate this barrier, the uptake to introduce technology-enabled teaching in infrastructure sustainability is still at a primary stage. Moreover, a lack of financial and in-kind support from employers to assist graduates to acquire specialised sustainability competencies is also considered as a major barrier. Often, fresh graduates require additional time and, in certain cases, financial assistance to be able to study in a specialised expertise area, such as sustainability. In particular, with the construction industry's stringent completion deadlines and complex working environments, it is challenging for fresh graduates to dedicate any additional time to further studies. Therefore, additional boosts, such as paid leave or extra funding allocations, are required at the organization level to motivate and enable graduates. However, organizations need to investigate policies and processes of retaining skilled graduates once the training is completed; otherwise, the rate of returns will be affected.

### 6.3. Requirement of an Integrated Approach towards CPD

The appointment of a dedicated role to lead the sustainability drive in a project is a recent trend in the majority of infrastructure projects. Acknowledging the importance of having a sustainability lead, either at the project level or the organization level, for key environmental sustainability competencies among technical members within the project team, is vital for the effective implementation of sustainable practices. As discussed, there are advantages and disadvantages of both a stand-alone course and complete integration of sustainability within a curriculum. In addition, the majority of the units in CM and CE courses cover broader levels of sustainability and lack a depth of sustainability topics. Therefore, an integrated approach would be an ideal solution that could motivate fresh graduates in the industry and enable undergraduates to further their environmental sustainability education in the infrastructure construction sector.

Evidently, there are limitations associated with the structure and content of existing CPD programs in the building and construction industry despite their potential value in developing expertise and capability. Hence, in addition to rapid advancements in construction technologies and continuous changes in policy, a dynamic CPD program is necessary to build on the education of industry practitioners and keep them up-to-date with the latest developments [9]. The involvement of industry experts to facilitate the delivery of several CPD activities is equally vital in designing and supporting the delivery of effective CPD programs. This will provide a platform for the structured evaluation of learning outcomes and an opportunity for systematic environmental sustainability education in the infrastructure construction industry. A multidisciplinary approach to CPD is recognised as an effective platform to foster collaboration and knowledge sharing, providing participants with an integrated understanding of the many disciplines required to delivery green infrastructure construction projects sustainably [50]. In addition, a dynamic digitised platform that can visualize the current progress, future milestones, and vibrant feedback process and peer-to-peer interaction of the designed CPD program would further encourage active participation and collaboration. With the advancements in Industry 4.0, the construction sector is rapidly moving towards digitisation [51], and the implementation of sustainability

education can strongly reap benefits by incorporating digitised collaborative learning platforms. This could eventually pave the way to a significantly upskilled workforce delivering environmentally sustainable infrastructure construction projects.

*6.4. Curriculum Development and Implementation*

Work-related experience is important in effectively conveying sustainability concepts in construction curricula. This is often achieved through work-integrated learning components in units such as guest lectures and site visits. The successful implementation of these learning components depends on creativity and practical relevance to environmental sustainability and can usually be achieved through problem-based learning. However, with a limited number of experts in both infrastructure construction and sustainability, it is often difficult to regularly source and integrate industry expertise into course delivery. Moreover, due to the unique and dynamic nature of different infrastructure project types, it is difficult to encompass all the sustainability aspects into one assessment. Due to these challenges, the horizontal integration of sustainability concepts within the curriculum seems to be complicated. The attraction for a stand-alone sustainability course at the undergraduate or postgraduate level specific to the infrastructure construction sector would be also quite limited as there will be fewer job demands for such a graduate. Therefore, designing several higher education curricula (at the subject level) with various pathways and catering to different stakeholders with vertical integration are necessary to improve the efficiency of the outputs.

## 7. Conclusions and Future Recommendations

In this paper, a three-step methodology was presented to assess the limitations of the environmental sustainability competencies in the Australian infrastructure construction sector. The first step in the three-step methodology involved a bibliometric analysis to identify research trends and gaps. It was found that Australian researchers have been less focused on the environmental sustainability education of infrastructure projects in civil engineering curricula, when compared to the building and construction curricula. The second step included an academic literature review of the following three sub-topics: (i) barriers to sustainable construction practices; (ii) environmental sustainability in built environment programs; and (iii) Continuous Professional Development (CPD). Finally, as the third step, a detailed analysis of the civil engineering and construction management undergraduate programs at Victoria University, Melbourne, was carried out to assess environmental sustainability education at an undergraduate level. Triangulating the findings from the three analyses yielded the identification of key issues to be addressed for the Australian infrastructure sector to successfully meet future sustainability targets. These included an integrated and structured approach towards CPD, effective curriculum development in undergraduate courses, financial support from government and industry for sustainability educational programs' delivery, and incentives and resources for graduates to pursue sustainability knowledge.

The current study used an Australian-based case study to investigate the extent of environmental sustainability considered in construction management and civil engineering curricula. These findings are case specific and can change from institution to institution, as well as from country to country based on the scope, objective, and focus of the course content related to environmental sustainability in transport infrastructure. However, the findings and the importance of different delivery strategies to promote environmental sustainability among construction and civil engineering professionals involved infrastructure construction projects. The general results can be effectively used to customize each curriculum within CM and CE courses at different institutions in various countries focusing on environmental sustainability. Future research should be focused on developing new learning components and delivery outputs for the environmental sustainability education of infrastructure construction and validate the findings through stakeholder participation. Future research can also be focused on using machine learning to develop a database that

can reach wider audiences within the higher education sector across the globe to highlight the importance of systematic sustainable education in the infrastructure sector. The bibliometric analysis highlighted weak advancements in technology-enabled education strategies to promote sustainable/green infrastructure. Therefore, the investigation of effective ways to use smart technologies such as Mixed Reality (MR) and Building Information Modelling (BIM) to facilitate the teaching and delivery strategies within higher education environmental sustainability curricula is important.

**Author Contributions:** Conceptualization, M.S.; methodology, M.S.; investigation, M.S. and Y.B.; writing—original draft preparation, M.S. and Y.B.; writing—review and editing, Z.V.; project administration, Z.V.; funding acquisition, Z.V. All authors have read and agreed to the published version of the manuscript.

**Funding:** This research was funded by the Victorian Higher Education State Investment Fund.

**Institutional Review Board Statement:** Not available.

**Informed Consent Statement:** Not available.

**Acknowledgments:** We gratefully acknowledge the State Government of Victoria, Australia, for providing financial support for this project.

**Conflicts of Interest:** The authors declare no conflict of interest.

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
