# Peer review of "Environmental Sustainability in Infrastructure Construction—A Review Study on Australian Higher Education Program Offerings"

_infrastructures, doi:10.3390/infrastructures7090109_

Round 1

Reviewer 1 Report

This study conducted a systematic review on Australian higher education for sustainability in the construction industry with a focus on infrastructure projects. It was interesting to assess limitations of the environmental sustainability competencies in the Australian infrastructure construction sector. This research can be improved to answer the comments listed below.

1.      This research focused on Australian higher education for sustainability. In order to engage with the wider readership, this paper’s contribution to the existing international literature should be addressed.

2.      What is the research gap that this study addressed, and what are contributions of this research?

3.      This study examined a three-step analysis, it should be explained that how these three steps are related.

4.      In the manuscript, ‘green infrastructure’ was mentioned several times. As this research focused on construction sector, how did authors define green infrastructure.

5.      In Figure 5, what are different sizes of circle mean?

6.      Australian Qualifications Framework (AQF) should be explained more. Table 4 showed matrix representation of learning outcome verbs corresponding to AQF levels and learning area. What are differences among the levels presented? How were the CM and CE curriculum analyzed?

7.       Figure 8 should be addressed in the manuscript. It is hard to understand each image mean.

8.      What are limitations of this research? Also, based on your study future study directions should be addressed.

Reviewer 2 Report

Dear Editor-in-Chief:

Journal: Infrastructures (ISSN 2412-3811)

Manuscript ID :infrastructures-1786769

This manuscript presented the Environmental sustainability in infrastructure projects – A review study on Australian higher education program offerings. The paper is worth publishing however, the following concerns should be addressed in revising the manuscript:

Introduction:

1.      The introduction is good, and you may need to cite additional references.

2.      It is preferable to clarify the goal of the current study more in proportion to the importance of the study.

Research Methodology:

1.      Figure 2 The definition of the x and y axis should be clarified

2.      Figure 3 The definition of the x and y axis should be clarified

3.      Figure 6  not clear

Conclusions and future recommendations

1.      It is preferable to put a special section on future recommendations

Thanks

Reviewer 3 Report

From my point of view, the authors have carried out a complete research project about an important and interesting issue. However, in my opinion, this manuscript is not acceptable to be published yet due to the following:

- The manuscript has several confusing and unclear parts and contents.

Some specific related comments are as follows:

- The presentation of the methodology is confusing and contradicts other parts of the manuscript. The reviewer has finally understood the methodology after reading the first paragraph of the conclusion. If this is the correct version of this project methodology, then the explanation of the methodology from section 2 must be improved. For instance, a first general explanation of the three steps is required at the beginning of section 2 - before explaining the details of the bibliometric assessment. Moreover, the number of steps must coincide, and Figure 1 must explicitly indicate and name these steps the same way that in the text.

- Similarly, the reviewer considers that the title has some confusing parts, mainly the first part and the word “projects”. Furthermore, the abstract also lacks concordance with the methodology and the discussion and conclusions sections, especially the main findings and future works.

- The reviewer does not understand the relation between the future works presented in the two last sentences of the conclusions and this research project. If the authors consider that these future works are clearly related to the research project they should further explain its relation.

Other comments:

- This article uses many abbreviations and some parts are difficult to follow. The reviewer advices the authors to do some improvements to ease the understanding of the article in this sense such as adding in Figure 8 and Table 5 a legend with the meaning of the abbreviations.

- Is figure 6 necessary?

- A last review of the spelling is necessary, i.e. “distriubtion" in Figure 7 caption.

Author Response

The authors would like to show their gratitude to the reviewer for taking time and effort to provide critical and constructive criticism on the manuscript. The authors are delighted to notice the recommendation of reviewer 3 of the worthiness of the manuscript and appropriateness for publication in the Infrastructures journal. The following section provides a clear description of the responses to the reviewers’ comments.

No

Comment

Response

Location

1.      From my point of view, the authors have carried out a complete research project about an important and interesting issue. However, in my opinion, this manuscript is not acceptable to be published yet due to the following:

2.      - The manuscript has several confusing and unclear parts and contents.

3.      Some specific related comments are as follows:

N/A

-

1

Introduction:

The presentation of the methodology is confusing and contradicts other parts of the manuscript. The reviewer has finally understood the methodology after reading the first paragraph of the conclusion. If this is the correct version of this project methodology, then the explanation of the methodology from section 2 must be improved. For instance, a first general explanation of the three steps is required at the beginning of section 2 - before explaining the details of the bibliometric assessment. Moreover, the number of steps must coincide, and Figure 1 must explicitly indicate and name these steps the same way that in the text.

Thank you for this comment.

The authors have revised Figure 1 and section 2 is revised to enhance the meaning of the three-step methodology.

 Figure 1 and Section 2

Similarly, the reviewer considers that the title has some confusing parts, mainly the first part and the word “projects”. Furthermore, the abstract also lacks concordance with the methodology and the discussion and conclusions sections, especially the main findings and future works.

The “projects” term in the title is replaced with “construction” to enhance the intended study scope in the current study.

The abstract has been revised to better coincide with the methodology and key findings.

Title

Abstract

The reviewer does not understand the relation between the future works presented in the two last sentences of the conclusions and this research project. If the authors consider that these future works are clearly related to the research project they should further explain its relation.

The future works related to technology enabled teaching and education is linked with the bibliometric findings, barriers and then recommended as a future works.

This is explained in the revised manuscript

Page 6-7 section 3

Section 6.2 page 22

Other comments:

This article uses many abbreviations and some parts are difficult to follow. The reviewer advices the authors to do some improvements to ease the understanding of the article in this sense such as adding in Figure 8 and Table 5 a legend with the meaning of the abbreviations.

A detailed explanation for Figure 8 is available on pg. 7. The abbreviations in Table 5 (unit codes) have been defined in Table 3. Hence, the authors believe a legend is not required in this Table.

N/A

- Is figure 6 necessary?

According to the authors’ knowledge, figure 6 was used to understand specific gaps in knowledge and education strategies related to green infrastructure in majority of the countries. To highlight the significance of Figure 6 an additional description is added into the revised manuscript.

Page 6-7 section 3

- A last review of the spelling is necessary, i.e. “distriubtion" in Figure 7 caption.

The entire manuscript is proof-read to correct typos, and grammar mistakes.

Entire manuscript

Round 2

Reviewer 1 Report

I think that all comments have been taken into account and the manuscript has been revised accordingly.

Author Response

The authors would like to acknowledge the reviewer’s efforts to check the submitted 1st revision of the manuscript after addressing the comments of the reviewers.

Based on the feedback provided, it is clear that the reviewer is satisfied with the content of the revised manuscript. 

Reviewer 3 Report

From my point of view, the authors have carried out a complete research project about an important and interesting issue. This version has improved compared to the previous version and is almost acceptable to be published though the following:

- This article uses many abbreviations and some parts are difficult to follow. The reviewer advices the authors to do some improvements to ease the understanding of the article in this sense such as adding in Figure 8 and Table 5 a legend with the meaning of the abbreviations. The reviewer has a lot of difficulties understanding these figures and foresees readers will have the same problem. Are the authors confident that any reader from outside their university will understand Figure 8?

Author Response

As per the reviewer’s suggestion, Figure 8 and table 5 has been updated in the revised manuscript.
